# Five Level Triage vs. Four Level Triage in a Quaternary Emergency Department: National Analysis on Waiting Time, Validity, and Crowding—The CREONTE (Crowding and RE-Organization National TriagE) Study Group

**DOI:** 10.3390/medicina59040781

**Published:** 2023-04-17

**Authors:** Gabriele Savioli, Iride Francesca Ceresa, Maria Antonietta Bressan, Gaia Bavestrello Piccini, Angelica Varesi, Viola Novelli, Alba Muzzi, Sara Cutti, Giovanni Ricevuti, Ciro Esposito, Antonio Voza, Antonio Desai, Yaroslava Longhitano, Angela Saviano, Andrea Piccioni, Fabio Piccolella, Abdel Bellou, Christian Zanza, Enrico Oddone

**Affiliations:** 1Department of Emergency Medicine and Surgery, IRCCS Fondanzione Policlinico San Matteo, 27100 Pavia, Italy; 2Department of Emergency, Ospedale Civile, 27029 Vigevano, Italy; 3Faculty of Medicine, University of Pavia, 27100 Pavia, Italy; 4Health Department, University of Pavia, 27100 Pavia, Italy; 5Department of Drug Science, University of Pavia, 27100 Pavia, Italy; 6Nephrology and Dialysis Unit, ICS Maugeri, University of Pavia, 27100 Pavia, Italy; 7Emergency Department, Humanitas University, Via Rita Levi Montalcini 4, 20089 Milan, Italy; 8Department of Biomedical Sciences, Humanitas University, Via Rita Levi Montalcini 4, Pieve Emanuele, 20072 Milan, Italy; 9Department of Anesthesia and Intensive Care, IRCCS Humanitas Research Hospital, Via Manzoni 56, Rozzano, 20089 Milan, Italy; 10Department of Anesthesiology and Intensive Care Medicine—AON Antonio, Biagio e Cesare Arrigo, 15100 Alessandria, Italy; 11Emergency Department, Fondazione Policlinico Universitario A. Gemelli, IRCCS, 00168 Roma, Italy; 12Institute of Sciences in Emergency Medicine, Guangdong Provincial People’s Hospital (Guangdong Academy of Medical Sciences), Southern Medical University, Guangzhou 510080, China; 13Department of Emergency Medicine, Wayne State University School of Medicine, Detroit, MI 48201, USA; 14Department of Public Health, Experimental and Forensic Medicine, IRCCS Fondazione Policlinico San Matteo, 27100 Pavia, Italy

**Keywords:** triage–emergency service, hospital, crowding, triage (under-triage), triage (over-triage), five level triage, four level triage, triage system, triage validity, waiting time, overcrowding and access block, overcrowding detection, overcrowding effect, overcrowding

## Abstract

*Background and Objectives:* Triage systems help provide the right care at the right time for patients presenting to emergency departments (EDs). Triage systems are generally used to subdivide patients into three to five categories according to the system used, and their performance must be carefully monitored to ensure the best care for patients. *Materials and Methods:* We examined ED accesses in the context of 4-level (4LT) and 5-level triage systems (5LT), implemented from 1 January 2014 to 31 December 2020. This study assessed the effects of a 5LT on wait times and under-triage (UT) and over-triage (OT). We also examined how 5LT and 4LT systems reflected actual patient acuity by correlating triage codes with severity codes at discharge. Other outcomes included the impact of crowding indices and 5LT system function during the COVID-19 pandemic in the study populations. *Results:* We evaluated 423,257 ED presentations. Visits to the ED by more fragile and seriously ill individuals increased, with a progressive increase in crowding. The length of stay (LOS), exit block, boarding, and processing times increased, reflecting a net raise in throughput and output factors, with a consequent lengthening of wait times. The decreased UT trend was observed after implementing the 5LT system. Conversely, a slight rise in OT was reported, although this did not affect the medium-high-intensity care area. *Conclusions:* Introducing a 5LT improved ED performance and patient care.

## 1. Introduction

Modern intra-hospital triage involves selecting and evaluating patients upon their arrival to an emergency department (ED) to determine their clinical status, diagnoses, and severity ratings to prioritize treatment access. Triage aims to place patients in the right area at the right time for the most appropriate treatment and to distribute medical resources according to the needs of patients. Trained triage nurses usually assign a triage level to patients using an established triage system. These systems seek to promote safe and efficient utilization of ED resources [1,2].

Triage systems recommend a “time to treatment” in EDs and measure service quality. These systems aim to standardize, create reproducible evaluation procedures, and regulate patient access to increasingly crowded EDs. Standardization increases patient safety and ED access equity by ensuring the quality of care for the community [3,4,5,6,7,8].

In EDs organized by areas of care intensity, patients are prioritized and channeled toward low- or medium-high-intensity care areas at triage [9,10]. Worldwide, triage models are distinguished mainly by the number of priority codes. A 3-level triage system (3LT) defines three priority codes for medical examination; there are also 4-level (4LT) and 5-level (5LT) triage systems [8,11,12].

In the 1980s, the number of 3- and 4LTs increased [8,11,12,13,14]. Gerald Fitzgerald introduced Australia’s first 5LT in 1986 [15]. This system, originally called the National Triage Scale, became the Australian Triage Scale (ATS) in 2000. The ATS demonstrated a superior correlation with patients’ acuity and adequate inter- and intra-observer reproducibility [3,13,16,17,18,19,20,21,22]. Three other 5LT systems followed in the 1990s: the Manchester Triage Scale (MTS), Canadian Triage and Acuity Scale (CTAS), and Emergency Severity Index (ESI) [23,24].

5LTs are considered a “gold standard” for their greater validity and reliability compared to 3LT or 4LT systems. Reliability, i.e., the degree of agreement on code assignment, has been demonstrated both inter and intra-observer: inter-rater agreement and intra-rater agreement. Validity, the ability to correlate with the true acuity of the patient, requires indirect indicators: such as correlation with the need for hospitalization, hospitalization in an intensive environment, the study of over- and under-triage, mortality [3,7,8,12,16,17,18,19,20,23,24,25]. The implementation of 5LT shows a clear correlation between the categories and indirect indices such as time to treatment, resource usage, intra-hospital mortality, hospitalization index, transfer frequency, and time spent in intensive care [3,8,11,12,16,17,18,19,20]. Triage systems also improve estimates of resource usage, hospital costs, the likelihood of hospitalization, and the risk of short-term mortality [17,18,26,27,28,29,30,31,32,33,34,35,36].

With 56.9% of hospitals responding, the most commonly used triage system in the USA is the 5LT ESI, followed by 3LT systems (25.2%) [37]. Unpublished 2021 data from SIMEU (Italian Scientific Society of Emergency Medicine) show that, in Italy, 40% of EDs use 5LT, 57% use 4LT, and 3% use 3LT systems [38]. 3LT and 4LT systems should be upgraded where possible.

In 2012, professionals and scientific societies gathered at the National Triage Coordination Conference and proposed a coding system with five priority codes. After a review of the literature on triage, in 2015, our research group began using a 5LT system that considers each patient’s symptoms, vital signs, and necessary treatment resources, as with other 5LTs [11,13,15,39].

Progressive increases in ED crowding are negatively linked to patient outcomes and satisfaction. However, crowding effects on triage wait times and the frequency of under- (UT) and over-triage (OT) are unclear and under-investigated. In addition, few studies have examined outcomes associated with 5LT systems and ED management in real-life settings [40,41,42].

In this special issue, we focus on ED management, re-engineering the triage system, and the influence of crowding in triage.

## 2. Methods

### 2.1. Study Design

This observational study was based on a retrospective review of the epidemiologic and clinical records of patients who visited the Foundation IRCCS Policlinic San Matteo from 1 January 2014 to 31 December 2020. We analyzed all ED visits that occurred during the 4LT period, from the inauguration of the new ED arranged by care intensity (from 1 January 2014 to 30 November 2015). These were compared to ED visits that occurred during the 5LT period (from 30 November 2015 to 31 December 2020).

Before 1 January 2014, our ED had a smaller layout, fewer resources, and no divisions by care intensity. We, therefore, do not believe that comparisons can be made with the period before 1 January 2014.

The data were extracted using PiEsse software, used to manage patients in our ED. Upon presentation, blood tests, imaging, and consultations are required; based on the results of these examinations, patients are hospitalized or discharged. We estimated changes in UT, OT, and crowding indices. The San Matteo Hospital Foundation provided data on all ED services. An ad hoc query was performed to obtain the data of interest. The patients’ names were anonymized to ensure confidentiality. All patients consented to have their data used for medical and research purposes and health data processing upon arrival to the ED, as required by the local ethics committee.

### 2.2. Endpoints

This analysis was carried out on the total ED accesses during two periods: 4LT and 5LT. We sought to determine the effects of introducing a 5LT on wait times. The secondary aim was to evaluate the impact of introducing a 5LT on validity, measured as UT, and OT. We also sought to determine if the 5LT system codes were better correlated (in comparison to 4LT system codes) to patients’ actual acuity; we verified this outcome by measuring the correlation between the triage code and severity code at discharge.

Other outcomes included the impact of triage on crowding indices such as the length of ED stay, total access block time, and rate of access block. Finally, we analyzed the functioning of the 5LT during the COVID-19 pandemic.

### 2.3. Inclusion and Exclusion Criteria

All non-pediatric patients (>14 years old) who visited the ED during the study periods were eligible for inclusion.

### 2.4. Study Population

Demographic data (sex and age), vital parameters (blood pressure, heart rate, oxygen saturation, Glasgow Coma Scale score, respiratory rate), signs and symptoms, waiting time, length of stay (LOS) in the ED, mode of ED access, priority codes for medical examination, exit codes for severity, total access block time, and rate of access block were collected for each patient. All medical records, including computed tomography data, were thoroughly reviewed. All collected data were entered into Microsoft Excel and used for statistical analysis.

The 5LT group consisted of 307,198 patients who accessed the ED between 30 November 2015 and 31 December 2020. The 4LT group consisted of 116,060 patients who accessed the ED during the control period between 1 January 2014 (inauguration of the new ED organized into areas of the care intensity) and 30 November 2015.

### 2.5. The Organizational Experience of Our ED

Since 2010, triage in our ED has been carried out with guided grids to determine triage codes (guided code attribution algorithms). These grids provide a calculated code but allow the operator to indicate the most appropriate code based on the overall patient assessment. These internal protocols were revised in 2015, according to Ministerial Recommendation No. 15, to adapt the triage activity to a new ED structure (organized by care intensity) and to implement a 5LT system. In 2019, at the same time the ED director retired, various internal reorganizations occurred. These included a shift from 7 to 12 h for doctors and nurses and the dissolution of the intensive short observation (OBI) team. Our region’s healthcare system is organized as a “hub-and-spoke” model. The terms “hub” and “spoke” are borrowed from the airline industry, where the “hub” represents the airport where most flights are concentrated. In medicine, the hub-and-spoke model assumes that, for certain complex pathologies, specialist practitioners and expensive equipment are needed. These resources cannot be guaranteed at every location. Consequently, more-complex patients are routed to regional or macro-area Centers of Excellence (“hubs”) for treatment. Peripheral medical centers (“spokes”) provide fewer and less-specialized services; therefore, less-complex patients are routed to these facilities. The hub-and-spoke model improves healthcare service provision [43].

#### Care Areas of the ED

Since December 2013, our ED has been divided into low- and medium-high-intensity areas. In addition to examination and shock rooms, the medium-intensity care area also includes an observation unit that functions as a holding area. The medium-high-intensity care area consists of a resuscitation area for triage code 1 (4 beds) and a level 2 critical monitoring area (6 beds). There is also a trauma area for more stable patients with fractures [9,10].

Incoming patients are first triaged by specialized nurses with basic and advanced triage education training. First, the nurses gather information about the main symptom(s) that led the patient to the ED, including a brief medical history. Second, they collect vital parameters and perform a visual inspection. Patients are then assigned a priority code for medical examination and are redirected toward a specific care intensity area. The triage process is based on written protocols (“triage grids” drawn up mainly based on the evolution of the main symptom) as well as the patient’s medical history and vital parameters.

Patients assigned to the medium-high-intensity area should demonstrate impairment of a vital parameter, an altered state of consciousness, be at increased evolutionary risk of any symptoms (e.g., typical chest pain), need specialized care (such as non-invasive ventilation), or multi-parameter monitoring. Once in the ED, the doctor establishes the patient’s diagnostic and therapeutic pathways. The two areas of the care intensity flow to a stabilization area for OBI (Figure 1).

### 2.6. The Two Triage Systems Are 4LT and 5LT Grids

The priority codes for medical examination in our ED are shown in the graph below. A triage code (the new Code 3) was introduced for patients requiring faster medical examination or therapy than other patients in the low-intensity care area. The secondary reason for the new Code 3 was to relieve patients who would have previously been assigned a Code 2 (area of medium intensity) because they needed urgent medical examination but did not require a medium-high-intensity care area (Figure 2).

In transitioning from a 4LT to a 5LT system, the triage codes for the medium-high care intensity remained unchanged: Code 1 indicates life-threatening conditions, and Code 2 indicates very urgent conditions. In the 5LT system, patients with Codes 3, 4, or 5 are allocated to the low-intensity care area. Previously, in the 4LT system, there were only two levels of triage code (Codes 3 and 4). Thus, an extra triage code was created for patients destined for the low-intensity area. Patients previously assigned to the triage code 2, who urgently needed to be seen by the doctor (high visit priority) but could otherwise be treated in a low-intensity care area, were assigned to this additional level of triage (e.g., pain due to renal colic).

### 2.7. UT and OT Definitions

The UT rate is defined as the proportion of patients who have a lower-than-appropriate triage code, are not attended to in time, or are not assigned to an adequate intensive care area. UT poses a serious risk and is associated with increased mortality and adverse outcomes. Calculation of UT using a two-entry contingency table corresponds to a false negative rate. These patients experienced low triage activation (assigned a triage code lower than necessary, assignment to an intensive care area lower than necessary, with reduced resource allocation) relative to all patients with that condition.

The OT rate is defined as the proportion of unnecessarily applied hospital resources to patients without a particular condition (resource overuse). Therefore, the calculation of OT using a two-entry contingency table corresponds to False Discovery Rate (FDR) = 1 − Positive Predictive Value. That is, it represents the ratio between those who had a high triage activation (assigned a triage code higher than necessary, assignment to an intensive care area higher than necessary; with an increased resource overuse) but did not have acute pathology compared to all those who had a high triage activation (both those with and without acute pathology). Figure 3 describes how the UT and OT are calculated.

### 2.8. Measurement of Crowding

Several widely validated indices for measuring crowding have been proposed [44,45,46,47,48,49,50,51,52,53,54,55,56,57,58,59,60,61,62,63,64,65,66,67]. The most used indices are:

Input crowding indices: wait times, number of patients visiting the ED, disease severity and complexity (e.g., number of patients at each acuity level), and number of people who left without being seen (LWBS).

#### Throughput Crowding Indices: LOS

Output crowding indices: mean number or percentage of admissions, patients in the ED (number or percentage), access block and boarding (mean number or percentage of patients who have experienced it), and access block or boarding times (such as the total access block time).

“Wait time” was defined as the total time from initial registration or triage to the time the patient was first seen by a doctor. The overall ED-LOS was the time from arrival at triage or registration until discharge or hospital transfer. LOS reflected the total patient experience, including care and waiting. Access block was defined as >8 h in the ED from presentation to admission [68]. Total access block time thus represented access block duration [69].

Boarding was the time that elapsed between the medical decision to admit the patient to the hospital and the patient’s arrival at the hospital bed; a boarding time in the ED of 6 h is considered high [70,71,72]. Thus, boarding time represented the duration of boarding [64,66,73,74].

We used indices that concerned both inputs and throughputs and output factors to account for the complexity of the crowding phenomenon. Among these we have chosen waiting time; ED-LOS; boarding and access block time.

### 2.9. Statistics

Continuous variables are described as means, medians, and interquartile ranges; qualitative variables are expressed as the number of observations and appropriate proportions. Between-group comparisons for continuous variables were made using the non-parametric Mann–Whitney test, according to their non-normal distributions. Associations between the qualitative variables were studied using the χ^2^ test. Statistical analyses were conducted using appropriate logistic multivariate regression models to test the association between time variables while accounting for crowding, exit block, and the different triage periods. The test of proportions was used to examine the differences in UT and OT by year of observation. In particular, for each record the presence/absence of over-triage and under-triage was modeled as a dichotomous variable, as described in the Methods section, and the risk of undergoing to either over- or under-triage was described as the odds ratio (OR) resulting from multiple regression analysis adjusted by age, gender and year of observation. This analysis has been carried out for the whole population and for subgroups in which boarding or exit block was present. The significance level was set at alpha 0.05 (statistical significance at *p* < 0.05), and all tests were two-tailed.

The analyzes were conducted with STATA software (version 14; Stata Corporation, College Station, TX, USA, 2015). The ethics committee submitted and approved the study (Protocol number 20200114609). The analyses were made on data from the PIESSE software (Piesse SRL, Latina, Italy).

## 3. Results

### 3.1. Overall (Table 1)

In the 5LT system period, there was a statistically significant increase in older patients (*p* < 0.001). During the same period, the number of patients that arrived at our ED by personal transport (so-called ambulatory arrivals) decreased significantly (39.1% vs. 20.8%, *p* < 0.001). Simultaneously, more patients presented by ambulance (27.6% vs. 37.2%, *p* < 0.001), and required specialized nursing staff (30.5% vs. 39.1%, *p* < 0.001) and medical assistance (2.5% vs. 2.7%, *p* < 0.001). A progressive increase in patients requiring higher triage codes was also reported (*p* < 0.001). The need for hospitalization progressively increased (32.6% vs. 55.5%; *p* < 0.001), transfers to spoke hospitals decreased (2.8% vs. 2.1%, *p* < 0.001), and the number of patients discharged decreased (64.2% vs. 42.1%, *p* < 0.001). Analyzing the correlation based on the care intensity in the two periods, the correlation between the triage code and the severity code at hospital discharge increased from 0.266 in 4LT to 0.319 in 5LT for the low-medium-intensity care area. For the medium-high care intensity area, the correlation improved from 0.277 in 4LT to 0.304 in 5LT.
medicina-59-00781-t001_Table 1Table 1(**a**) Principal personal and ED presentation features of patients included in the study, by period of observation. The 4LT period (T4) spanned 1 January 2014 to 30 November 2015; the 5LT period (T5) spanned 1 December 2015 to 31 December 2020. ^a^: χ^2^ test. (**b**) Pathology at admission for the patients included in the study, by period of observation.**(a)**
1-1-2014/30-11-20151-12-2015/31-12-2020
Sex4LT*n* (%)5LT*n* (%)p ^a^Male59,432 (51.2)158,914 (51.7)
Female56,628 (48.8)148,283 (48.3)0.002Age


<1811,333 (9.8)27,267 (8.9)
18–2915,975 (13.8)39,128 (12.7)
30–3913,711 (11.8)32,049 (10.4)
40–4916,376 (14.1)40,943 (13.3)
50–5914,132 (12.2)41,339 (13.5)
60–6912,698 (10.9)34,955 (11.4)
70–7914,902 (12.8)41,001 (13.4)
80+16,933 (14.6)50,516 (16.4)<0.001Triage priority code


Code 513,443 (11.6)25,748 (8.4)
Code 478,777 (67.9)191,981 (62.5)
Code 30 (-)17,297 (5.6)
Code 222,711 (19.6)67,688 (22.0)
Code 11129 (0.9)4484 (1.5)<0.001Priority code at discharge


Code 529,240 (25.2)43,141 (14.0)
Code 473,995 (63.8)224,039 (72.9)
Code 30 (-)425 (0.1)
Code 211,952 (10.3)36,341 (11.8)
Code 1873 (0.7)3252 (1.2)<0.001Care intensity


Low92,220 (79.5)235,026 (76.5)
Medium-to-high23,840 (20.5)72,172 (23.5)<0.001Outcome


Discharge94,701 (81.6)246,413 (80.2)
Hospitalization17,347 (14.9)51,043 (16.6)
Transfer2166 (1.9)5746 (1.9)
Left without being seen1385 (1.2)2933 (0.9)
Other461 (0.4)1063 (0.4)<0.001**(b)**
**Pathology at ED Access*****n*****%**4TL



Trauma39,71334.22
Major trauma2710.23
Minor symptoms25,61422.07
Dyspnea53994.65
Thoracic pain58705.06
Abdominal pain94558.15
Headache43533.75
Neurologic symptoms16301.40
Bleeding20241.74
Fever/Sepsis10.00
Other28,481118.735TL



Trauma12,2335.03
Major trauma9330.38
Minor symptoms35,71214.70
Dyspnea14,1175.81
Thoracic pain17,3217.13
Abdominal pain26,15910.76
Headache34941.44
Neurologic symptoms14,3195.89
Bleeding57572.37
Fever/Sepsis80483.31
Other10,492443.18


### 3.2. Wait Time (Table 2)

There was a minimal reduction in wait times for life-threatening triage codes (5 min for Code 1 patients during the 4LT system period vs. 4.3 min during the 5LT system period, *p* < 0.001). In contrast, wait times rose for very urgent codes (23.5 min for Code 2 patients during the 4LT system period vs. 32.5 min during the 5LT system period, *p* < 0.001). Comparing the twelve months prior to the introduction of the 5LT with the twelve months that followed 5LT implementation (10,636 cases in T4 and 13,608 in T5), we noted a minimal, non-significant increase of ~3 min. Considering triage code 2 data acquired over various years, we saw a slight (non-significant) yet constant increase in wait times of ~3–4 min per year, corresponding with the increase in the number of triage code 2 patients and crowding at our hospital. More precisely, the year our facility switched from 4LT to 5LT, there was a greater increase in just less than 4 min. This increase continued during the 4LT (just under 4 min from the year 2014–2015) and 5LT (just under 2 min from 2016 to 2017; about 5 min from 2017 to 2018) periods. The wait time increased significantly in 2019.

The wait times for Code 3 in the 5LT system period were similar to those of Code 2 in the 4LT system period (24.3 min for Code 3 during the 5LT system period vs. 23.5 min for Code 2 patients during the 4LT system period). The wait times for Codes 4 and 5 during the 5LT system period were comparable to those of Codes 3 and 4 during the 4LT system period (52.1 min for Code 3 patients during the 4LT system period vs. 57.5 min for Code 4 patients during the 5LT system period and 52.2 min, for Code 4 patients during the 4LT system period vs. 48.4 min for Code 5 patients during the 5LT system period).
medicina-59-00781-t002_Table 2Table 2(**a1**) Selected time variables accounting for crowding, by period. * The 4LT period (T4) spanned 1 January 2014 to 30 November 2015; the 5LT period (T5) spanned 1 December 2015 to 31 December 2020; ^a^: Kruskal–Wallis test. (**a2**) Wait time, by period and code at presentation. * The 4LT period (T4) spanned 1 January 2014 to 30 November 2015; the 5LT period (T5) spanned 1 December 2015 to 31 December 2020. ^a^: Kruskal–Wallis test. (**b**) Selected time variables accounting for crowding, by presence of boarding and exit block. ^a^: Kruskal–Wallis test. (**c**) Wait time (Mean; minutes) for triage code 2 the 12 months before, and the 12 months immediately following, the introduction of the 5LT system. * Kruskal–Wallis test. (**d**) Wait time (Mean; minutes) for triage code 2 during the seven years of the study. The 4LT period (T4) spanned 1 January 2014 to 30 November 2015; the 5LT period (T5) spanned 1 December 2015 to 31 December 2020.**(a1)**
**Period *****Observations****Median (min)*****p*****^a^****Interquartile Range (min)**Wait time





T4116,06043.3
16.7–96.1
T5307,19845.5<0.00117.7–104.5Process time





T4116,060105.7
52.1–194.1
T5307,198118.4<0.00157.5–232.2Length of stay (LOS)





T4116,060174.2
99.0–290.8
T5307,198195.8<0.001108.2–338.1**(a2)**
**Period *****Observations****Median (min)****Interquartile Range (min)*****p*****^a^**Wait time




Code 5





T413,44352.218.2–109.1

T525,74848.417.5–104.3<0.001Code 4





T478,77752.120.7–108.9

T5191,98157.522.3–122.9<0.001Code 3





T517,29724.312.9–44.9-Code 2





T422,71123.511.4–49.1

T567,68832.514.2–73.8<0.001Code1





T411295.02.6–9.8

T544844.32.2–8.5<0.001**(b)**


**Observations****Median (min)****Interquartile Range (min)*****p*****^a^**Wait timeLow-intensity care






No boarding28,73152.721.5–114.7


Boarding741662.424.8–141.1<0.001
Medium-to-high care intensity






No boarding35,22518.78.0–44.3


Boarding493023.19.9–54.6<0.001
Low-intensity care






No exit block29,00548.620.3–105.4


Exit block714294.334.3–186.5<0.001
Medium-to-high care intensity






No exit block35,90718.48.0–43.1


Exit block424828.211.5–71.2<0.001**(c)**
**N****Wait Time****(Median; min)****Interquartile Range*****p********4LT10,63625.312.3–52.9
5LT13,60828.113.2–58.20.001**(d)****Year**
**N****Mean****Median****Interquartile Range**2014T412,07536.222.010.7–46.4
T5----2015T410,63640.025.312.3–52.9
T5101143.828.914.5–60.92016T4----
T512,59743.728.013.1–58.02017T4---

T513,26345.528.312.8–60.62018T4----
T514,57652.733.514.5–72.92019T4----
T514,52570.943.617.2–103.52020T4----
T511,71658.532.713.8–80.1


### 3.3. UT and OT 

The risk of UT tended to decrease in the 5LT compared to the 4LT system period (Table 3; OR = 0.87, *p* < 0.001). Table 3 shows trends relative to care intensity areas. The three conditions most represented by this phenomenon (UT in medium-high-intensity) were chest pain (23.5%), dyspnea (22.6%), and neurological disorders (20.1%). The most frequent causes of these three symptoms were non-ST-elevation myocardial infarction (STEMI), acute heart failure, pneumonia, and stroke. The trend of the phenomenon (UT in medium-high-intensity) seems to have fluctuated over the years (2014, *n* = 135, 1.12%; 2015, *n* = 175, 1.50%; 2016, *n* = 248, 1.97%; 2017, *n* = 212, 1.60%; 2018, *n* = 181, 1.24%; 2019, *n* = 188, 1.29%; 2020, *n* = 210, 1.79%). UT has decreased since 2016, reaching a nadir in 2018 and gradually increasing again in 2019 and 2020 (Table 4).

The risk of OT tended to slightly increase during the 5LT period compared to the 4LT period (Table 3; OR = 1.16; *p* < 0.001). However, OT increases in the medium-high care intensity area were minimal and non-significant (Table 3; OR = 1.05, *p* = 0.03).

### 3.4. Crowding

Boarding and exit blocks observed from 2014 to 2020 indicated that crowding had progressively increased (Table 4). The number of ED visits rose gradually until 2018, except for a slight deflection in 2015, and then decreased in 2019 and 2020 (Table 4). Boarding and exit blocks indicated longer wait times for low and medium-high-intensity care areas (Table 2; *p* < 0.001). We have chosen the crowding indices that are most reproducible with an automated data extraction [13,75,76,77,78,79]. Boarding substantially reduced the risk of OT and slightly decreased the risk of UT in both care intensity areas, as shown in Table 5a. Exit block substantially reduced the risk of OT and slightly reduced the risk of UT, as shown in Table 5b.

### 3.5. LT of COVID Patients

Our ED encountered 3826 patients with COVID-19. Of these, 125 received a triage code 5, 2789 received a triage code 4, 169 received a triage code 3, 810 received a triage code 2, and 86 received a triage code 1. Of all of these, there were 159 positives for COVID-19 at a PCR test, and 78 positives died in the ED. The main wait times in this area were, respectively, 48 min for Code 5, 47 min for Code 4; 48 min for Code 3; 27 min for Code 2, and 10 min for Code 1.

## 4. Discussion

### 4.1. Overall

This study analyzed the validity of a 5LT over a 4LT. To our best knowledge, this is the first published Italian study, the largest European study, and the first study conducted on an ED organized by care intensity area [80]. Although some real-life 5T studies have been performed, most of these have been performed on a single symptom or disease, often on a small number of patients. Some of these were wait time studies, and others were real-life 5LT validation studies [81,82,83,84,85].

Our work analyzes the function of triage by capturing the complexity of real life and studies the functionality and impact on waiting times over a long period for all causes of access. It is the first Italian study that compares the validity of 4LT and 5LT through the calculation of UT and OT on a large population sample. 

Triage validity is the ability of the triage system to correlate with the patient’s acuity; since the real acuity of the patient is impossible to detect, surrogate indices are used such as the following: UT and OT, correlation with hospitalizations, mortality and hospitalization in resuscitation [86,87].

This is, to our knowledge, the first study that analyzes the mutual influences between triage and crowding by studying all the determinants of crowding, and in particular its main determinants: boarding and exit blocks.

Patients who present to the ED have become more numerous, frail, and sicker. This is underlined by the progressive rise in the patients’ ages, the decrease in spontaneous accesses, the higher number of accesses through the territorial emergency service or on a stretcher, and the number of higher triage and severity codes at discharge. This phenomenon has been reported in the literature for several years [44,45,46,47,49,50,51,52,53,54,55,56,64,65,73,74,75,88], which has increased since the onset of the COVID-19 pandemic and translated into higher hospitalization rates [76,77]. These factors increase exit block and boarding, which, together with the progressive reduction in available hospitalization beds, worsens crowding.

Collectively, these factors have changed the way ED physicians work, transforming their practice from “admit-to-care” to “care-to-admit” [13,76,77,78,79]. This change is reflected in a gradual extension of LOS and process times. Because of these changes, treatment and observation paths have been designed for pathologies that are also very complex or with complex management, such as, for example, severe trauma, heart failure and head trauma. [89,90,91,92,93,94,95]. Although increased crowding negatively affects patient outcomes and satisfaction, the consequences for triage—particularly on wait times and UT and OT frequency—have not yet been extensively investigated. Improvements in the correlation between the triage code and the severity code at discharge suggest that 5LT system triage codes more accurately reflect actual patient acuity compared to the 4LT system. These data, interpreted alongside data indicating a general reduction in UT, show how reducing the risk of UT directly benefits sicker patients, potentially improving their subsequent outcomes.

### 4.2. Waiting Time

In transitioning from a 4LT to a 5LT system, the triage codes for the medium-high care intensity remained unchanged: Code 1 for life-threatening conditions and Code 2 for very urgent conditions. In the 5LT system, patients with Codes 3, 4, or 5 were allocated to the low-intensity care area. Previously, in the 4LT system, there were only two levels of triage code (Code 3 and 4). Thus, an extra triage code was created for patients destined for the low-intensity area. Patients previously assigned triage code 2, who urgently needed to be seen by the doctor (high visit priority) but could be treated in a low-intensity care area, were assigned a Code 3 (e.g., pain due to renal colic).

The additional triage level, which characterized the implementation from the 4LT system to the 5LT system, enhanced our ability to identify patients in the low-intensity area who urgently needed to be seen by a doctor. Thus, it is expected that the benefits or the disadvantages of the 5LT introduction will be more evident in the low-intensity area.

The new Code 3 aimed to identify the most fragile or compromised patients to guarantee them the best possible path without inappropriate use of medium or high care intensity areas.

Codes 4 and 5 patients did not experience significantly increased wait times, underscoring one benefit of the 5LT system. However, though the wait times for Code 1 patients improved slightly, those for Code 2 patients lengthened. Because this could not be attributed to the transition to the 5LT system, the increased wait time for Code 2 was constant over the years before and after the shift to the 5LT system and corresponded with the increased presentation of Code 2 patients and hospital crowding. The increase that occurred in 2019 against a slight reduction in the number of Code 2 patients seems to be related to increased crowding that year during ED reorganization following a change in leadership. The 2020 drop seems related to the COVID-19 pandemic, characterized by a collapse of input factors and the exposure of throughput and output factors.

Exit block and boarding most frequently affect patients requiring admission or transfer to another health facility and are prevalent in higher care intensity ED areas. In addition to the examination and shock rooms, the medium-intensity care area also includes a holding area [9,10]. The medium-high-intensity care patient needs oxygen, non-invasive ventilation, telemetry, and continuous monitoring of vital parameters. These are, in turn, dependent on structural limits (i.e., the number of oxygen outlets, ventilators, and monitors). Increased patient assignments to this area and a simultaneous rise in the exit block result in resource saturation, with consequently higher times needed for patient processing.

Other changes in the internal departmental organization may also have contributed to increased LOS (i.e., doctor and nurse turn-over, different structure of shifts, etc.). However, they were analyzed by this study. As highlighted by Zoubir et al. [79], the association between boarding time and ED outcomes is still under investigation; Multicenter studies are needed to better clarify the various effects of boarding and exit block on adverse ED outcomes.

Additional attention should be paid to the performance of doctors and nurses working in triage. Placement of triage physicians allows for rapid disposition of low-acuity patients. Meanwhile, more-complex patients can be evaluated sooner, somewhat mitigating the effects of ED crowding. Placement of senior doctors, defined as a medical doctor who completed high specialty training in emergency medicine, with nurses in triage improves wait time, LOS, LWBS rates, and left without treatment complete rates. Furthermore, medical-nurse triage teams hold advantages for direct admission of medically complex or frail patients (e.g., elderly patients) who do not require time-dependent interventions but require hospitalization. Triage care teams can also reduce overcrowding [96,97,98].

### 4.3. UT and OT 

The introduction of 5LT systems has reduced UT. Similar results have been shown in studies that analyzed the transition from 3- or 4LT to 5LT systems [22,80,99]. These results demonstrate that the 5LT systems are safer than 4LT systems: reducing UT similarly reduces unfavorable patient outcomes. Remarkably, this risk reduction is most evident in the low-intensity care area, which was also the most changed by introducing the 5LT system. The trend toward an increase in medium-high-intensity UT was already present during the 4LT period; however, the phenomenon clearly demonstrates an oscillatory trend. It, therefore, seems reasonable not to attribute this phenomenon to the 5LT system.

Conversely, some symptoms (chest pain, dyspnea, and acute neurological disorders) have a major role in the studied period. UT can be attributed to the rapid evolution of these symptoms, with possible serious and unpredictable impairment of the patient’s clinical evolution. Examples may be ECG changes in the examination room that could lead to the activation of the STEMI pathway, sudden deterioration to coma, or shock resulting from these symptoms.

In our ED, UT tended to decline after the introduction of the 5LT system, reaching its nadir in 2018. Subsequent increases were observed in 2019 and 2020, with an increase in exit block and boarding phenomena. Notably, in these two years, the worsening of crowding outputs and throughput factors corresponded to a reduction in the number of ED visits.

While the 2020 trend could have resulted from the COVID-19 pandemic, the 2019 trend was likely influenced by the internal reorganization of our ED following a change of leadership, as already described in 2.5.

As described for some diseases, the reduction in UT is accompanied by an increase in OT [100,101]. Although we observed a slight increase in OT, it did not affect the medium-high care intensity area. Analysis of the OT trend over the years shows three periods of reductions in conjunction with reductions in the number of ED visits. This suggests that OT might be influenced by the number of ED visits. It is possible that, when visits increase, the tacit habit of “better to over- than to under-triage” can prevail, as previously assumed by some authors [102].

### 4.4. Crowding Indices

The data analyzed in this study underline the progressive increases in crowding. The rise in LOS, exit block, boarding, and processing times have resulted in a net increase in throughput and output factors, with a consequent lengthening of wait times. The increase is higher with exit blocks than boarding in low-intensity areas and for less-urgent triage codes. The presence of an exit block almost doubled the wait time. Furthermore, wait times were lengthened by approximately 25–30% for patients with high-priority codes who required medical examination (triage Codes 1 or 2).

These data demonstrate the influence of output factors on ED pathways. The exit block increases processing and LOS times, with a consequent slowdown of all ED processes and flows, including wait and handling times. The situation can be likened to a funnel into which patients continuously pour and whose outflow is limited by the diameter of the neck. On one side of the funnel is the “city”, on the other (the narrow one) the hospital with “beds”. The tighter the neck, the slower patient flow, starting at triage (Figure 4).

With the low-intensity care and lower priority triage codes, both boarding and exit blocks substantially reduce the risk of OT and slightly reduce the risk of UT. Moreover, in both cases, triage is more accurate for patients with lower triage priorities. This may be due to increased attention from triage nurses in cases of crowding. However, the greater accuracy may also depend on the long wait times when patients are re-evaluated more easily and frequently.

In patients with life-threatening or urgent conditions (triage Codes 1 and 2), UT is reduced in boarding and, more significantly, in cases of exit block. In contrast, while OT in these patients is essentially unchanged with boarding, it increases in the event of an exit block. There is also an increased risk of OT when there are more ED visits. This phenomenon could be due to greater attention being paid by the triage nurses in cases of crowding, reflecting improved accuracy during boarding and exit block.

When facing overcrowding, attention tends to focus on reducing UT, which is even more reduced than in boarding alone (this time at the expense of increased OT).

Perhaps the worst accuracy, determined only by an increase in OT, might be because patients with Codes 1 and 2 also have longer wait times than usual; however, these wait times were not long enough to allow easier and more frequent re-evaluations. Therefore, it would seem that in the event of a large increase in crowding, overestimation is preferred to avoid UT. The response of “better to over- than to under-triage” could contribute to this effect. This phenomenon would be more valid during urgent visits and with patients requiring medium-high-intensity care.

Finally, it should be noted that the COVID-19 epidemic began in 2020. As shown in Table 4, crowding doubled in 2020, straining the system. This highlighted crowding effects on triage. Thus, 5LT confirmed the abovementioned advantages in the context of a pandemic.

### 4.5. The 5-Level Triage in COVID Patients

The present study includes the first year of the COVID outbreak. Our center has been involved since the dawn of the epidemic being one of the HUB centers for COVID in north Italy and having treated patient zero. During the COVID-19 pandemic, the ED underwent a profound reorganization to mitigate the risk of contagion and streamline access for patients based on need. During the first pandemic phase of 2020, an area was created in the Infectious Diseases building for positive COVID patients. Those who required hospitalization were referred to specialty inpatient wards for positive patients. When the pressure of the pandemic decreased, this need was maintained, and a similar area was created inside the general ED. Parallel to this, we began prohibiting visitors and companions (except for special circumstances) to reduce the risk of virus transmission. Entrance point screening with a SARS-CoV-2 nasal swab and rapid (within 6 h) results reporting was critically important for identifying positive patients. 

Patients arriving with the territorial emergency service or independently, and presenting fever, respiratory symptoms or COVID-like symptoms were referred to COVID triage. With this process, 3826 patients were then referred to the “COVID flow” in the year 2020 and were then subjected to COVID triage. All patients already identified positive at other centers and transferred only for treatment directly to dedicated COVID wards are therefore excluded from our database. Waiting times in the dedicated COVID area were in-line with general times.

Patients were treated and stabilized, and before being sent to the wards or being discharged, they had to wait for the result of a molecular PCR test. Those who were positive were referred to hospital wards dedicated to COVID patients. Patients with a negative test and with low clinical suspicion with an alternative diagnosis were referred to COVID-free wards. Patients with negative tests whose reports instead leaned toward a COVID infection were transferred to a gray area where they would repeat the tests.

Of these molecular tests performed in ED, only 159 were positive. The mortality of these patients remained high, while the management times were comparable to patients in the other areas.

### 4.6. Strengths and Limitations of the Study

The large size of our study cohort is certainly a strength of this study. However, this study was also limited by its retrospective design. We believe that simulated results are important; however, even well-designed studies will depart from “real-life” data obtained over a vast period. This study allowed us to analyze real-world processes in a “real” medically complex clinical cohort. Our conclusions are limited because of the study’s observational nature, including retrospective retrieval of information. Another limitation is that the study is monocentric one.

## 5. Conclusions

(i) The introduction of the 5LT has proved beneficial for the management of waiting times compared to the 4LT.

(ii) The introduction of a 5LT reduced UT, with a contextual increase in OT. These trends were directly proportional to the number of ED visits and increased crowding due to the worsening of output factors (such as access block) and the increase in input factors (such as the number of ED visits). Future research is needed to find models to reduce the increase in OT while trying not to lose the benefit of UT. 

(iii) Crowding indices, such as boarding and access block, are related to increased wait times. The increase in boarding and access block over the years in fact corresponds to an increase in waiting times. Future research must be oriented toward finding and verifying response models that, by measuring the output factors in real time, can activate adequate triage responses aimed at limiting this worsening. This response must take into account that in the event of an increase in crowding, a reduction in the UT and a possible increase in the OT is expected.

(iv) The increase in crowding indices during the COVID pandemic underlines how this situation requires specific answers, on the design and feasibility of which future research will have to focus.

Triage remains an open challenge for the emergency physician even with 5LT introduction. Implementing AI triage algorithms in nursing routine could overcome age-specific issues. These data suggest that all Italian EDs should consider transitioning to the 5LT model.

## Figures and Tables

**Figure 1 medicina-59-00781-f001:**
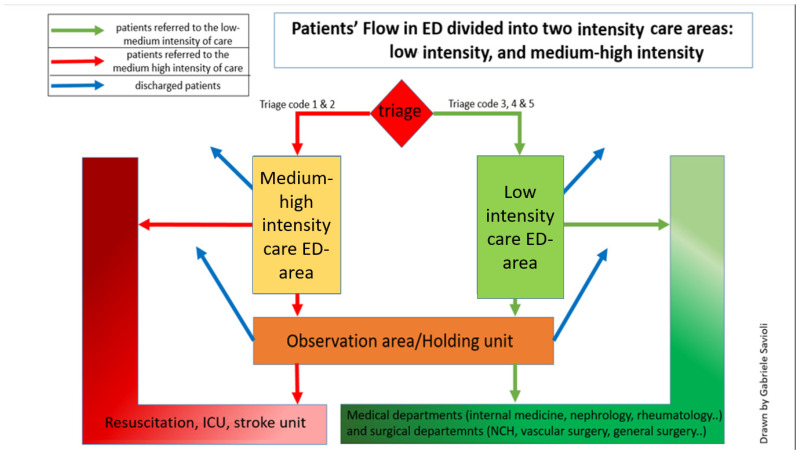
Our ED is divided into two areas according to care intensity. A medium-high-intensity area (shown on the right in the figure) includes a shock room for cases to be isolated and medium-high-intensity beds. Patients at high developmental risk and requiring high care intensity (codes 1 and 2) are routed here. In a second area, patients with low-intensity care are managed, and some patients with medium-intensity care (codes 3, 4, and 5) can be managed. The two areas are physically separated while remaining connected through two corridors so that low- and high-intensity patient flows do not cross. Patients can be sent to the observation unit—where patients are stabilized, monitored, and observed—from both areas. Our observation unit also functions as a holding unit from the observation unit; patients can be hospitalized, transferred to other hospitals, or discharged. Admission and discharge are directly possible from both areas.

**Figure 2 medicina-59-00781-f002:**
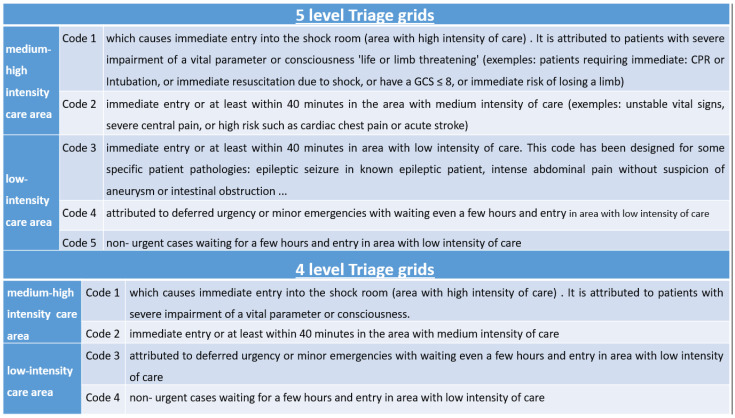
4LT and 5LT grids.

**Figure 3 medicina-59-00781-f003:**
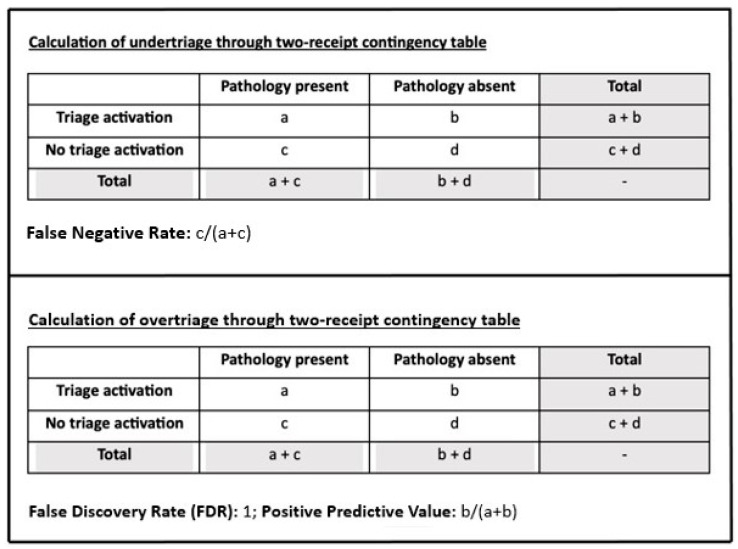
Calculation of UT and OT.

**Figure 4 medicina-59-00781-f004:**
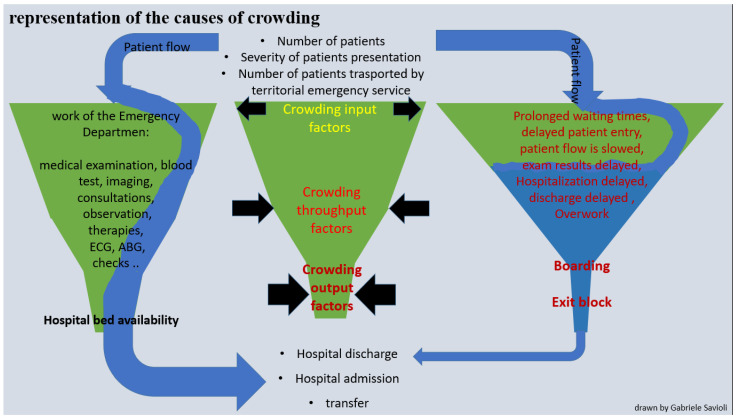
Graphic representation of the causes of crowding. This figure represents crowding in EDs. The ED is represented by a funnel. The volume of patients who present to the ED is represented by the water which enters the funnel (blue arrow). The input factors (number of incoming patients, number of serious incoming codes, number of patients arriving by ambulance) are a large part of the funnel input. The throughput factors (blood tests, imaging, instrumental tests, consultations, checks, number of staff on medical and nursing shifts, tight shifts) comprise the body of the funnel. The output factors (exit block, boarding) are represented by the neck of the funnel. In a normal situation (left column), the flow of patients (blue arrow) enters the ED (the funnel) and leaves after normal processing (medical examination performed, any blood, any imaging, any consultations). The times, imaginatively represented by the time required for water to flow through the funnel, are normal in this situation. The central column represents crowding or increases in input factors, as in the case of hyper-influx or simultaneous arrival of medically complex or critically ill patients (situation represented by an enlarged funnel base), or due to internal factors, such as presentation of medically complex patients who require prolonged stabilization or numerous medical procedures (as represented by an enlarged funnel body) or for the worsening of the outgoing factors, as is necessary in the case of exit block (situation represented in this case by a restricted funnel neck). The resulting situation (right column) sees a global and marked slowdown in patient flow (blue arrow) and prolongation of time points (waiting, process, LOS). Normally, the outgoing flow is wider. In cases of crowding, it is markedly slowed, as represented by a thinner blue arrow at the exit.

**Table 3 medicina-59-00781-t003:** Risk of UT and OT by period.

	**Period ***	**OR ^a^**	**95% Confidence Interval**	** *p* **
Over-triage				
Low-intensity care	4LT	1.00 (Ref.)	-	
	5LT	-	-	-
Moderate-to-high-intensity care	4LT	1.00 (Ref.)	-	
	5LT	1.05	1.01–1.11	0.03
Total	4LT	1.00 (Ref.)	-	
	5LT	1.16	1.14–1.19	<0.001
Under-triage				
Low-intensity care	4LT	1.00 (Ref.)	-	
	5LT	0.85	0.82–0.88	<0.001
Moderate-to-high-intensity care	4LT	1.00 (Ref.)	-	
	5LT	1.35	1.12–1.65	0.002
Total	4LT	1.00 (Ref.)	-	
	5LT	0.87	0.84–0.91	<0.001
	**Period ***	**OR ^a^**	**95% Confidence Interval**	** *p* **
Over-triage				
Low-intensity care	4LT	1.00 (Ref.)	-	
	5LT	-	-	-
Moderate-to-high-intensity care	4LT	1.00 (Ref.)	-	
	5LT	1.03	1.00–1.07	0.07
Total	4LT	1.00 (Ref.)	-	
	5LT	1.08	1.04–1.12	<0.001
Under-triage				
Low-intensity care	4LT	1.00 (Ref.)	-	
	5LT	1.05	1.03–1.08	<0.001
Moderate-to-high-intensity care	4LT	1.00 (Ref.)	-	
	5LT	1.18	1.03–1.34	0.014
Total	4LT	1.00 (Ref.)	-	
	5LT	1.03	1.01–1.06	0.019

* The 4LT period (T4) spanned 1 January 2014 to 30 November 2015; the 5LT period (T5) spanned 1 December 2015 to 31 December 2020. ^a^: Odds ratios (OR) estimated by multiple regression analysis adjusted by age, sex, and year of observation.

**Table 4 medicina-59-00781-t004:** (**a**) ^#^ Proportion of boarding and exit block (calculated only on hospitalized patients) from 2014 to 2020; (**b**) Table + Figure. Evolution of boarding from 2014 to 2020; (**c**) Table + Figure. Evolution of Over Triage (OT) from 2014 to 2020; (**d**) Table + Figure. Evolution of Under Triage (UT) from 2014 to 2020.

**(a)**								
	**2014**	**2015**	**2016**	**2017**	**2018**	**2019**	**2020**	** *p* ** **for Trend**
Boarding ^#^								
No	9404	9030	9617	9792	10,041	8781	7291	
	91.0%	89.9%	88.6%	87.2%	87.2%	81.2%	63.3%	
Yes	926	1010	1241	1431	1475	2033	4230	
	9.0%	10.1%	11.4%	12.8%	12.8%	18.8%	36.7%	<0.001
Exit Block ^#^								
No	9544	9089	9717	9934	10,148	8792	7688	
	92.4%	90.5%	89.5%	88.5%	88.1%	81.3%	66.7%	
Yes	786	951	1141	1289	1368	2022	3833	
	7.6%	9.5%	10.5%	11.5%	11.9%	18.7%	33.3%	<0.001
Accesses per day	165.8	165.3	170.8	174.4	176.8	175.8	129.8	
Number of accesses	60,512	60,336	62,527	63,662	64,540	64,181	47,500	
**(b)**								
**Year**	**Number of Patients with Boarding**		**Number of Hospitalized Patients**	**Proportion A**		**Total Number of Patients**		**Proportion B**
2014	926		10,330	0.098		60,512		0.016
2015	1010		10,040	0.112		60,336		0.017
2016	1241		10,858	0.129		62,527		0.020
2017	1431		11,223	0.146		63,662		0.023
2018	1475		11,516	0.147		64,540		0.023
2019	2033		10,814	0.232		64,181		0.033
2020	4230		11,521	0.580		47,500		0.098
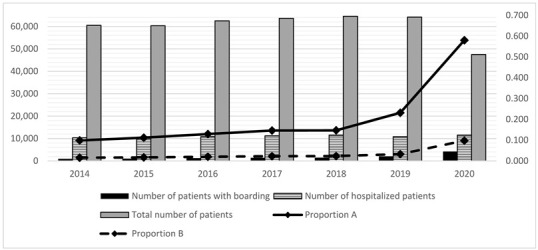								
**(c)**								
**Year**	**Number of Patients with OT**			**Total Number of Patients**				**Proportion**
2014	8006			60,512				0.132
2015	7553			60,336				0.125
2016	8550			62,527				0.137
2017	9097			63,662				0.143
2018	10,396			64,540				0.162
2019	10,198			64,181				0.159
2020	7339			47,500				0.155
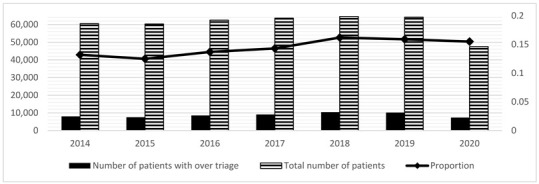								
**(d)**								
**Year**	**Number of Patients with UT**			**Total Number of Patients**				**Proportion**
2014	4806			55,706				0.079
2015	4952			55,384				0.082
2016	5069			57,458				0.081
2017	5065			58,597				0.080
2018	4890			59,650				0.076
2019	5000			58,181				0.079
2020	5143			42,357				0.108
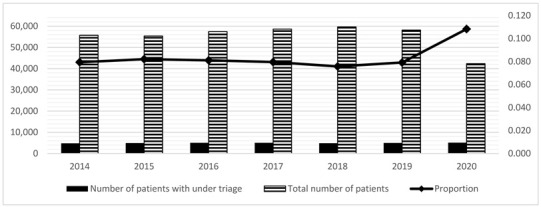								

**Table 5 medicina-59-00781-t005:** (**a**) Risk of UT and OT, by presence of boarding. (**b**) Risk of UT and OT, by presence of exit block.

**(a)**				
	**Boarding**	**OR ^a^**	**95% Confidence Interval**	** *p* **
Over-triage				
Low-intensity care	No	1.00 (Ref.)	-	
	Yes	0.60	0.07–5.20	0.641
Moderate-to-high-intensity care	No	1.00 (Ref.)	-	
	Yes	0.98	0.91–1.05	0.576
Total	No	1.00 (Ref.)	-	
	Yes	0.68	0.63–0.73	<0.001
Under-triage				
Low-intensity care	No	1.00 (Ref.)	-	
	Yes	0.92	0.88–0.98	0.004
Moderate-to-high-intensity care	No	1.00 (Ref.)	-	
	Yes	0.82	0.69–0.98	0.032
Total	No	1.00 (Ref.)	-	
	Yes	0.91	0.87–0.96	0.001
**(b)**				
	**Exit Block**	**OR ^a^**	**95% Confidence Interval**	** *p* **
Over-triage				
Low-intensity care	No	1.00 (Ref.)	-	
	Yes	0.65	0.07–5.60	0.691
Moderate-to-high-intensity care	No	1.00 (Ref.)	-	
	Yes	1.12	1.04–1.20	0.004
Total	No	1.00 (Ref.)	-	
	Yes	0.69	0.64–0.74	<0.001
Under-triage				
Low-intensity care	No	1.00 (Ref.)	-	
	Yes	0.92	0.87–0.97	0.004
Moderate-to-high-intensity care	No	1.00 (Ref.)	-	
	Yes	0.72	0.58–0.88	0.001
Total	No	1.00 (Ref.)	-	
	Yes	0.91	0.86–0.96	<0.001

^a^: Odds ratios (OR) estimated by multiple regression analysis adjusted by age, sex and calendar year of observation.

## Data Availability

The raw data supporting the conclusions of this article will be made available by the authors upon reasonable request.

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
