# Peer review of "Five Level Triage vs. Four Level Triage in a Quaternary Emergency Department: National Analysis on Waiting Time, Validity, and Crowding—The CREONTE (Crowding and RE-Organization National TriagE) Study Group"

_medicina, 2023, doi:10.3390/medicina59040781_

Round 1

Reviewer 1 Report

Various systems are used in emergency departments worldwide to assess the severity of illness of patients and to prioritise treatment (triage).

The emergency department of a clinic is the crucial interface between the ambulance service and the clinic. However, emergency departments are increasingly being chosen as the primary access point to healthcare, which is reflected in an annual increase in patient numbers. Since the patient volume in emergency rooms is difficult to plan, only some of the patients have life-threatening or medically urgent problems and not all patients can be treated immediately and simultaneously, emergency patients with life-threatening illnesses must be reliably identified within a few minutes at the time of presentation.

The sole recording of vital signs is not suitable for identifying critically ill patients in the emergency department. Internationally, different triage systems are therefore used for initial assessment. These include unstructured assessment according to the patient's own professional experience. Instruments such as a 3-step, 4-step or 5-step scale. Many studies confirm the high validity and reliability of the 5-step scale. Five-step triage systems are therefore recommended by national and international emergency medical societies.

The authors of the publication analysed precisely 7 years of the triage system, involving a four-stage as well as a five-stage system. The method has been very well defined. Priority code initial and at discharge, outcome, pathology at ED access have been compared.

A very important aspect is also the waiting time depending on the category, and in this article the authors also make a detailed comparison of the times in both systems. It is also interesting to provide based on the results a graphic the causes of crowding.

The work also has the benefit of looking at the extraordinary period as a pandemic, which required more careful work and focus on triage services.

Author Response

Various systems are used in emergency departments worldwide to assess the severity of illness of patients and to prioritise treatment (triage).

The emergency department of a clinic is the crucial interface between the ambulance service and the clinic. However, emergency departments are increasingly being chosen as the primary access point to healthcare, which is reflected in an annual increase in patient numbers. Since the patient volume in emergency rooms is difficult to plan, only some of the patients have life-threatening or medically urgent problems and not all patients can be treated immediately and simultaneously, emergency patients with life-threatening illnesses must be reliably identified within a few minutes at the time of presentation.

Dear Reviewer, we thank you for the appreciation of our work. We fully agree with you on the photograph you portray on the reality of emergency departments

The sole recording of vital signs is not suitable for identifying critically ill patients in the emergency department. Internationally, different triage systems are therefore used for initial assessment. These include unstructured assessment according to the patient's own professional experience. Instruments such as a 3-step, 4-step or 5-step scale. Many studies confirm the high validity and reliability of the 5-step scale. Five-step triage systems are therefore recommended by national and international emergency medical societies.

Dear Reviewer, we fully agree on what you expose of the various triage systems.

The authors of the publication analysed precisely 7 years of the triage system, involving a four-stage as well as a five-stage system. The method has been very well defined. Priority code initial and at discharge, outcome, pathology at ED access have been compared.

A very important aspect is also the waiting time depending on the category, and in this article the authors also make a detailed comparison of the times in both systems. It is also interesting to provide based on the results a graphic the causes of crowding.

The work also has the benefit of looking at the extraordinary period as a pandemic, which required more careful work and focus on triage services.

Dear Reviewer, thank you for fully summarizing our work by highlighting the most significant aspects.

Reviewer 2 Report

In this retrospective, observational study the authors assessed the effects of introducing a 5-level triage (5LT) system on waiting time, under- and overtiage, and crowding. They compared the data to the previously used 4-level triage (4LT) system. The topic is extremely relevant in today's era when most emergency departments are struggling with the same challenges: exit blocks, crowding and as a consequence: increased waiting times and higher risk of patients coming to harm.

The authors analysed an impressive number of presentations (over 423000!). The analysed time period for the 4LT system was almost 2 years, while for the 5LT it includes more than 6 years. The introduction gives an appropriate overview of the topic. 

The methods section is well-written. It is interesting that the 5LT group has almost three times more subjects than the 4LT group however, the reason behind this was explained in the "Study design" section.

Figure 1 shows "medium-high intensitive care" and "medium-low intensitive" areas. Were these supposed to be "intensive care" or "intensity care" areas? Please clarify. It is also confusing, that in the text the authors refer to "medium-high" and "low" intensity areas (not medium-high and medium-low). Could the authors be more consistent please?

Table 1.b "Principal personal and ED presentation features of patients included in the study" shows some major differences between the two cohorts. Namely Trauma (34.22% vs 5.03%), Fever/sepsis (0% vs 3.31%) and Other (18.73% vs 43.18%) Can the authors please provide and explanation of the differences? It is very unlikely that no patient attended the department with Fever/Sepsis during the 4LT period and it is really odd that the number of trauma cases dropped by this much during the 5LT phase. What is the reason behind these differences? Does this introduce any potential bias into the calculations? Are the two datasets even comparable if they have fundamental differences in the number of presenting pathologies?

In section 3.55. "LT of COVID patients": The authors mention that they included 3826 COVID patients, yet only 159 (!?) of them were tested positive for COVID-19. Can the authors please explain this discrepancy? What were the criteria for "COVID patient"? Which type of test was used? (PCR ? antigen?)

The section about crowding and it's correlation with UT/OT is well-written and thought-provoking. A topic that is difficult to assess considering the fundamental differences in the structure of various hospitals and ED departments. 

on line 555, please include the type of COVID test used. (I presume this was a PCR test? or a POC antibody test followed by PCR?)

In the conclusions section the authors mention that using Artificial Intelligence (AI) could help overcoming age-specific issues. What issues exactly? How would AI help in the triage algorithm? While I find the use of AI in triage a very exciting topic, mentioning it in the second to last sentence without any previous context seems to be a bit detached and irrelevant.

Please also correct the typos and the few grammatical errors in the text.

Author Response

In this retrospective, observational study the authors assessed the effects of introducing a 5-level triage (5LT) system on waiting time, under- and overtiage, and crowding. They compared the data to the previously used 4-level triage (4LT) system. The topic is extremely relevant in today's era when most emergency departments are struggling with the same challenges: exit blocks, crowding and as a consequence: increased waiting times and higher risk of patients coming to harm.

Dear Reviewer, we agree with you on the situation you describe of the emergency departments.

The authors analysed an impressive number of presentations (over 423000!). The analysed time period for the 4LT system was almost 2 years, while for the 5LT it includes more than 6 years. The introduction gives an appropriate overview of the topic. 

The methods section is well-written. It is interesting that the 5LT group has almost three times more subjects than the 4LT group however, the reason behind this was explained in the "Study design" section.

Dear Reviewer, we thank you for the plot shown for our work.

Figure 1 shows "medium-high intensitive care" and "medium-low intensitive" areas. Were these supposed to be "intensive care" or "intensity care" areas? Please clarify. It is also confusing, that in the text the authors refer to "medium-high" and "low" intensity areas (not medium-high and medium-low). Could the authors be more consistent please?

Dear reviewer, thank you very much for your comment that allows us to make this aspect of our work clearer.

We have corrected the figure: the areas of intensity of care are effectively divided into low intensity of care and medium-high intensity of care. We apologize for the TYPO in the figure

Table 1.b "Principal personal and ED presentation features of patients included in the study" shows some major differences between the two cohorts. Namely Trauma (34.22% vs 5.03%), Fever/sepsis (0% vs 3.31%) and Other (18.73% vs 43.18%) Can the authors please provide and explanation of the differences? It is very unlikely that no patient attended the department with Fever/Sepsis during the 4LT period and it is really odd that the number of trauma cases dropped by this much during the 5LT phase. What is the reason behind these differences? Does this introduce any potential bias into the calculations? Are the two datasets even comparable if they have fundamental differences in the number of presenting pathologies?

The encoding of access reasons has remained the same with the same program both in the years in which we have used the 4level triage system, and in the years in which we have used the 5LT. Therefore the two groups are comparable.

However, it is true that triage differences have been created over the years.

Since 2016, the regional major trauma network has been strengthened. Major trauma patients were then centralized at our HUB center for severe trauma with increasing frequency while trauma that did not meet major trauma criteria were referred to neighboring spoke centers. Hence the differences between severe trauma and trauma that have been created over the years.

As far as fever is concerned, on the other hand, going to see the diagnoses at discharge we actually realized that patients with infectious diagnoses (pneumonia, urinary tract infection ...) were framed at the beginning as minor symptoms, a category most represented in the 4LT. It was only with the introduction of 5LT and training courses that such patients were identified as fever and not as minor symptoms (22% 4LT vs 14% 5LT).

It is precisely this decrease in the minor symptoms category that can partly explain the variation in the Other category. First of all, it should be remembered that Other is not an existing category, but the set of all the other initial classification items that we have grouped for the purposes of the article in others, in order to avoid a very long list that we believe diverts the reader's attention. This category therefore includes, among others: lower back pain, back pain, heart pain, burns, wounds, epigastralgia. Headache, intoxication, psychiatric disorders..

The reduction in the use of the minor symptoms category has certainly contributed to the greater use of the other categories, with a consequent increase in the other category. Obviously, the wide range of the Other super-category can in itself justify variations of the same.

In section 3.55. "LT of COVID patients": The authors mention that they included 3826 COVID patients, yet only 159 (!?) of them were tested positive for COVID-19. Can the authors please explain this discrepancy? What were the criteria for "COVID patient"? Which type of test was used? (PCR ? antigen?)

Thank you for your comment that allows us to better clarify this aspect of our work.

The work includes the first year of the COVID outbreak. Our center has been involved since the dawn of the epidemic being one of the HUB centers for COVID in north Italy and having treated patient zero. Patients arriving or accompanied by the territorial emergency service or arriving independently and presenting fever, respiratory symptoms or COVID-like symptoms were referred to COVID triage. With this process, 3826 patients were then referred to the “COVID flow” in the year 2020, and were then subjected to COVID Triage. All patients already identified positive at other centers and transferred only  for treatment directly to dedicated COVID wards are therefore excluded from our database.

Patients there were treated and stabilized.

Before being sent to the wards or before being discharged, they were waiting for the result of a molecular PCR test. Those who were positive were referred to hospital wards dedicated to COVID patients. Patients with a negative test and with lower clinical suspicion because the investigations performed posed an alternative diagnosis were referred to COVID-free wards. Patients with negative tests whose reports instead leaned towards a COVID infection were transferred to a gray area where they would repeat the tests.

Of these molecular tests performed in ED, only 159 were positive.

We have included this information in the discussion

The section about crowding and it's correlation with UT/OT is well-written and thought-provoking. A topic that is difficult to assess considering the fundamental differences in the structure of various hospitals and ED departments. 

Dear Reviewer honored for your appreciation to this session

on line 555, please include the type of COVID test used. (I presume this was a PCR test? or a POC antibody test followed by PCR?)

Dear reviewer we have entered the data: we confirm it is a PCR molecular test.

In the conclusions section the authors mention that using Artificial Intelligence (AI) could help overcoming age-specific issues. What issues exactly? How would AI help in the triage algorithm? While I find the use of AI in triage a very exciting topic, mentioning it in the second to last sentence without any previous context seems to be a bit detached and irrelevant.

Dear reviewer, the enthusiasm of our sentence comes from another work we are preparing. However, we agree that it is superfluous in this context. Expanding the already rich work for a contribution that in this article would remain only theoretical could divert the reader from the true objectives of the present work. Therefore we agree to remove the sentence.

Please also correct the typos and the few grammatical errors in the text.

 Dear reviewer, we have done.

Reviewer 3 Report

Comments on: Evaluation of a real-life five-level triage system in a crowded emergency department during the COVID-19 pandemic in Italy: The CREONTE (Crowding and R E Organization National Triage) study

This paper addresses an interesting topic about the triage system in a crowded emergency department and concludes that the introduction of a 5LT system has improved emergency department performance and patient care.

Overall, I must say that this study is well-written and easy to read. There are, however, some points to be considered. Please see some comments and suggestions below.

The authors regarding the contribution of the paper to the literature state that: “Although other studies have analyzed the impact of 5LT models on wait times, UT, and OT, none have simultaneously examined crowding.” However, waiting times can also be understood as a measure of crowding, as the authors mention at some point in the paper. Therefore, I believe that the paper’s contribution to the literature should be clarified and improved, even if it may appear limited.

I am afraid the authors have mixed up two ideas. The idea of comparing the two systems 4TL and 5TL, whose data for the 4TL range from January 1, 2014, to November 30, 2015, and for the 5TL range from November 30, 2015, to December 31, 2020, with the idea of analysing the COVID. By this, I mean that despite the title referring to COVID, its analysis is very limited, which can be seen in section 3.55 of the paper.

Please clarify the discrepancy in the dates of December 31, 2020, in the abstract and December 31, 2021, in the methodology.

Please write out the full terms of UT and OT the first time they appear in the main text.

In Table 1b, please correct the number of observations of “other” in the 4TL.

In section 3.2, I suggest providing an association between the codes and the colors in Table 2.b to make interpretation easier. For example (page 9, line 295): “for Code 1 (Red Code) patients during…”

Please confirm if there was supposed to be a Table 2 or if was a formatting issue. Additionally, could you explain what samples the Mann-Whitney test and Kruskal-Wallis test concern? Is there any particular reason for using both tests?

Lastly, I have a question about Table 3. For example, the estimated odds ratio for over-triage related to “moderate to high-intensity care” is 1.05 in 5TL. Does this mean that patients with “moderate to high-intensity care” are more likely to be over-triaged in the 5TL period compared to the 4TL period? Additionally, does this estimation only apply to cases reported with “moderate to high-intensity care”? Right?

Author Response

Comments on: Evaluation of a real-life five-level triage system in a crowded emergency department during the COVID-19 pandemic in Italy: The CREONTE (Crowding and R E Organization National Triage) study

This paper addresses an interesting topic about the triage system in a crowded emergency department and concludes that the introduction of a 5LT system has improved emergency department performance and patient care.

Overall, I must say that this study is well-written and easy to read. There are, however, some points to be considered. Please see some comments and suggestions below.

Dear Reviewer, thank you for the appreciation you show for our work

The authors regarding the contribution of the paper to the literature state that: “Although other studies have analyzed the impact of 5LT models on wait times, UT, and OT, none have simultaneously examined crowding.” However, waiting times can also be understood as a measure of crowding, as the authors mention at some point in the paper. Therefore, I believe that the paper’s contribution to the literature should be clarified and improved, even if it may appear limited.

Dear reviewer, thank you for this comment that allows us to better highlight the contribution of our work. We propose to change the sentence to the following::

Although some real-life 5T studies have been performed, most of these have been performed on a single symptom or disease, often on a small number of patients. Some of these were wait time studies, other real-life 5LT validation studies..

Our work with its vast number analyzes the function of triage by capturing the complexity of real life. It studies the functionality and impact on waiting times over a long period for all causes of access. It is the first Italian study that appears the validity of 4LT and 5LT through the calculation of UT and OT on a large population.

Triage validity means its ability to correlate with the patient's acuity; since the real acuity of the patient is impossible to detect, surrogate indices are used such as: UT and OT, correlation with hospitalizations, mortality and hospitalization in resuscitation.

It is the first study, to our knowledge, that analyzes the mutual influences between triage and crowding by studying all the determinants of crowding, and in particular its main determinants: boarding and exit blocks.

I am afraid the authors have mixed up two ideas. The idea of comparing the two systems 4TL and 5TL, whose data for the 4TL range from January 1, 2014, to November 30, 2015, and for the 5TL range from November 30, 2015, to December 31, 2020, with the idea of analysing the COVID. By this, I mean that despite the title referring to COVID, its analysis is very limited, which can be seen in section 3.55 of the paper.

Dear reviewer, we agree that the title thus expressed is misleading, therefore we propose to modify it as follows:

Real-life comparison between 4 level triage and five level triage systems in Italy: waiting times, validity, crowding influence.

Please clarify the discrepancy in the dates of December 31, 2020, in the abstract and December 31, 2021, in the methodology.

We apologize for the TYPO: the correct date is December 31, 2020. We have corrected the text.

Please write out the full terms of UT and OT the first time they appear in the main text.

Dear review, we have provided.

In Table 1b, please correct the number of observations of “other” in the 4TL.

Dear review, we have provided.

In section 3.2, I suggest providing an association between the codes and the colors in Table 2.b to make interpretation easier. For example (page 9, line 295): “for Code 1 (Red Code) patients during…”

Dear review, we have provided.

Please confirm if there was supposed to be a Table 2 or if was a formatting issue. Additionally, could you explain what samples the Mann-Whitney test and Kruskal-Wallis test concern? Is there any particular reason for using both tests?

Dear reviewer, we apologize for the TYPO, in the formatting had in fact disappeared the 2nd table,

Table 2a. Selected time variables accounting for crowding, by period.    

Period*

Observations

Median (min)

pa

Interquartile range (min)

Wait time

T4

116,060

43.3

16.7 – 96.1

T5

307,198

45.5

<0.001

17.7 – 104.5

Process time

T4

116,060

105.7

52.1 – 194.1

T5

307,198

118.4

<0.001

57.5 – 232.2

Length of stay (LOS)

T4

116,060

174.2

99.0 – 290.8

T5

307,198

195.8

<0.001

108.2 – 338.1

* The 4LT period (T4) spanned 01/01/2014 to 30/11/2015; the 5LT period (T5) spanned 01/12/2015 to 31/12/2020.; b: Kruskal–Wallis test.

In the first version of our manuscript we included in tables both mean and median values (and related statistical tests) to provide more information to the readers. Following your suggestions, we dropped out the mean values (and related test) for the sake of simplicity and to not generate misunderstanding or confusion to the readers. We kept only median values and their interquartile range due the non-normal distribution of considered variables. Thus, we modified the Tables accordingly

Table 2.a – Wait time, by period and code at presentation.

Period*

Observations

Median (min)

Interquartile range (min)

pa

Wait time

Code 5

T4

13,443

52.2

18.2–109.1

T5

25,748

48.4

17.5–104.3

<0.001

Code 4

T4

78,777

52.1

20.7–108.9

T5

191,981

57.5

22.3–122.9

<0.001

Code 3

T5

17,297

24.3

12.9–44.9

-

Code 2

T4

22,711

23.5

11.4–49.1

T5

67,688

32.5

14.2–73.8

<0.001

Code1

T4

1,129

5.0

2.6–9.8

T5

4,484

4.3

2.2–8.5

<0.001

The 4LT period (T4) spanned 01/01/2014 to 30/11/2015; the 5LT period (T5) spanned 01/12/2015 to 31/12/2020. a: Kruskal–Wallis test.

Table 2.b - Selected time variables accounting for crowding, by presence of boarding and exit block.

Observations

Median (min)

Interquartile range (min)

pa

Wait time

Low-intensity care

No boarding

28,731

52.7

21.5–114.7

Boarding

7,416

62.4

24.8–141.1

<0.001

Medium-to-high care intensity

No boarding

35,225

18.7

8.0–44.3

Boarding

4,930

23.1

9.9–54.6

<0.001

Low-intensity care

No exit block

29,005

48.6

20.3–105.4

Exit block

7,142

94.3

34.3–186.5

<0.001

Medium-to-high care intensity

No exit block

35,907

18.4

8.0–43.1

Exit block

4,248

28.2

11.5–71.2

<0.001

a: Kruskal–Wallis test.

Table 2.c - Wait time (Mean; minutes) for triage code 2 the 12 months before, and the 12 months immediately following, the introduction of the 5LT system.

N

Wait time

(Median; min)

Interquartile range

p*

4LT

10636

25.3

12.3–52.9

5LT

13608

28.1

13.2–58.2

0.001

* Kruskal–Wallis test.

Table 2.d - Wait time (Mean; minutes) for triage code 2 during the seven years of the study.

Year

N

Mean

Median

Interquartile range

2014

T4

12075

36.2

22.0

10.7–46.4

T5

-

-

-

-

2015

T4

10636

40.0

25.3

12.3–52.9

T5

1011

43.8

28.9

14.5–60.9

2016

T4

-

-

-

-

T5

12597

43.7

28.0

13.1–58.0

2017

T4

-

-

-

T5

13263

45.5

28.3

12.8–60.6

2018

T4

-

-

-

-

T5

14576

52.7

33.5

14.5–72.9

2019

T4

-

-

-

-

T5

14525

70.9

43.6

17.2–103.5

2020

T4

-

-

-

-

T5

11716

58.5

32.7

13.8–80.1

The 4LT period (T4) spanned 01/01/2014 to 30/11/2015; the 5LT period (T5) spanned 01/12/2015 to 31/12/2020.

Lastly, I have a question about Table 3. For example, the estimated odds ratio for over-triage related to “moderate to high-intensity care” is 1.05 in 5TL. Does this mean that patients with “moderate to high-intensity care” are more likely to be over-triaged in the 5TL period compared to the 4TL period? Additionally, does this estimation only apply to cases reported with “moderate to high-intensity care”? Right?

Exactly. In the specific example, patients with "moderate to high-intensity care" are more likely to be over-triage in the 5TL period than in the 4TL period, with low statistical sigificativity (p=0.03); This estimate applies only to cases reported with "moderate to high-intensity care". The total estimate is then provided

Round 2

Reviewer 2 Report

The quality of the paper is greatly improved compared to the first version.

I only have a minor observation:

on Figure 1, the bottom of the top part is cut off

Reviewer 3 Report

In general, I found the paper’s content has improved to the point where it can be accepted for publication